# Controversies in the Science of Sedentary Behaviour and Health: Insights, Perspectives and Future Directions from the 2018 Queensland Sedentary Behaviour Think Tank

**DOI:** 10.3390/ijerph16234762

**Published:** 2019-11-27

**Authors:** Stuart J.H. Biddle, Jason A. Bennie, Katrien De Cocker, David Dunstan, Paul A. Gardiner, Genevieve N. Healy, Brigid Lynch, Neville Owen, Charlotte Brakenridge, Wendy Brown, Matthew Buman, Bronwyn Clark, Ing-Mari Dohrn, Mitch Duncan, Nicholas Gilson, Tracy Kolbe-Alexander, Toby Pavey, Natasha Reid, Corneel Vandelanotte, Ineke Vergeer, Grace E. Vincent

**Affiliations:** 1Institute for Resilient Regions, University of Southern Queensland, Springfield Central, QLD 4300, Australia; jason.bennie@usq.edu.au (J.A.B.); katrien.decocker@usq.edu.au (K.D.C.); tracy.kolbe-alexander@usq.edu.au (T.K.-A.); ineke.vergeer@usq.edu.au (I.V.); 2Baker Heart and Diabetes Institute, Melbourne, VIC 3004, Australia; david.dunstan@bakeridi.edu.au (D.D.); neville.owen@bakeridi.edu.au (N.O.); 3School of Public Health, The University of Queensland, Brisbane, QLD 4072, Australia; p.gardiner@uq.edu.au (P.A.G.); g.healy@sph.uq.edu.au (G.N.H.); b.clark3@uq.edu.au (B.C.); n.reid@uq.edu.au (N.R.); 4Cancer Council Victoria, Melbourne, VIC 3004, Australia; brigid.lynch@cancervic.org.au; 5The University of Queensland, RECOVER Injury Research Centre, Faculty of Health and Behavioural Sciences, Brisbane, QLD 4072, Australia; c.brakenridge@uq.edu.au; 6School of Human Movement and Nutrition Sciences, The University of Queensland, Brisbane, QLD 4072, Australia; wbrown@uq.edu.au (W.B.); n.gilson1@uq.edu.au (N.G.); 7College of Health Solutions, Arizona State University, Phoenix, AZ 85004, USA; mbuman@asu.edu; 8Department of Neurobiology, Care Sciences and Society, Karolinska Institute, 11330 Stockholm, Sweden; ing-mari.dohrn@ki.se; 9Faculty of Health and Medicine, University of Newcastle, Newcastle, NSW 2308, Australia; mitch.duncan@newcastle.edu.au; 10Faculty of Health, Queensland University of Technology, Brisbane, QLD 4001, Australia; toby.pavey@qut.edu.au; 11School of Health, Medical and Applied Sciences, Central Queensland University, Rockhampton, QLD 4702, Australia; c.vandelanotte@cqu.edu.au; 12School of Health, Medical and Applied Sciences, Central Queensland University, Wayville, SA 5034, Australia; g.vincent@cqu.edu.au

**Keywords:** breaks, debate, health, mediation, moderation, physical activity, posture, sedentary, standing

## Abstract

The development in research concerning sedentary behaviour has been rapid over the past two decades. This has led to the development of evidence and views that have become more advanced, diverse and, possibly, contentious. These include the effects of standing, the breaking up of prolonged sitting and the role of moderate-to-vigorous physical activity (MVPA) in the association between sedentary behaviour and health outcomes. The present aim is to report the views of experts (n = 21) brought together (one-day face-to-face meeting in 2018) to consider these issues and provide conclusions and recommendations for future work. Each topic was reviewed and presented by one expert followed by full group discussion, which was recorded, transcribed and analysed. The experts concluded that (a). standing may bring benefits that accrue from postural shifts. Prolonged (mainly static) standing and prolonged sitting are both bad for health; (b). ‘the best posture is the next posture’. Regularly breaking up of sitting with postural shifts and movement is vital; (c). health effects of prolonged sitting are evident even after controlling for MVPA, but high levels of MVPA can attenuate the deleterious effects of prolonged sitting depending on the health outcome of interest. Expert discussion addressed measurement, messaging and future directions.

## 1. Introduction

Sedentary behaviour has been defined as “any waking behaviour characterized by an energy expenditure ≤1.5 metabolic equivalents (METs), while in a sitting, reclining or lying posture”. It has been identified as a distinct behavioural entity from physical inactivity [1], which is the term used to reflect insufficient physical activity or exercise for health gains. Hence, an individual could have rather little ‘sitting’, due to their job, for example, but equally engage in a very low volume of health-enhancing moderate-to-vigorous physical activity (MVPA). Conversely, a highly active person (e.g., a daily runner) is clearly not inactive, but could engage in high amounts of sitting throughout the rest of the day.

The field of sedentary behaviour research expanded greatly from the early 2000s. Searches of SCOPUS to the end of 2018 reveal an exponential increase in papers with ‘sedentary’ and ‘sedentary behavio[u]r’ in the title (see Figure 1). With ‘sedentary behaviour’ probably better reflecting the literature of interest to the current paper, it can be seen that there has been a greater than 10-fold increase in the number of published papers over the past two decades. Epstein’s seminal laboratory studies with children were important papers concerning behavioural influences and changes in both physical activity and sedentary behaviours [2]. However, it was Salmon’s initial population-based findings showing relationships of TV time with body mass index (BMI) across physical activity strata, along with the review presented by Owen et al. of environmental determinants of physical activity and sedentary behaviour, that signalled important early contributions to the field in adults [3,4].

There is an even longer history, however, of systematic research on seated occupations and the use of adjustable workstations in ergonomics [5,6], effects of long-haul flights on conditions such as venous thromboembolism (deep vein thrombosis; DVT, with case studies reported in French as early as 1951) [7,8] and the health effects of seated leisure pursuits, such as TV viewing in children and adolescents [9], as well as the seminal occupational physical activity research by Morris and colleagues [10]. In the latter, seated occupations, such as bus driving, were compared with more active jobs in respect of health outcomes. While Morris’ work is considered to be the foundation of modern-day ‘physical activity epidemiology’, attention was primarily directed to the health effects of physical ‘inactivity’ rather than sitting per se. For example, it was concluded that “men in physically active jobs have a lower incidence of coronary heart disease in middle age than have men in physically inactive jobs” [10] p. 1120. However, it could be reasonably argued that such work was also influential, albeit much later, in considering the evident trend towards less physical activity at work and its replacement by ‘lighter’ moving or sedentary tasks. This is illustrated by declining energy expenditure in the occupational setting over the past five decades [11].

The development of sedentary behaviour research can be mapped onto the behavioural epidemiology framework [12]. The phases of this framework comprise measurement of the behaviour, the association between the behaviour and health, correlates of the behaviour, behaviour change through interventions and translational efforts to roll out behaviour change solutions. Considerable scientific endeavour has been directed towards each of these phases. For example, the measurement of sedentary behaviour has involved the development of self-reported measures that attempt to capture total sedentary time as well as sitting in different domains (e.g., work, leisure, transport) [13,14,15,16]. Moreover, developments in technology now allow for the assessment of posture and movement using wearable devices [17]. Extensive research efforts have been invested in testing associations between sedentary behaviour and multiple health outcomes (e.g., all-cause mortality, cardiovascular disease (CVD)). One of the first such studies showed that a simple five-category self-report (‘almost none of the time, one fourth of the time, half of the time, three fourths of the time, almost all of the time’) of sitting time was associated with mortality in a large population sample of Canadians. Such an association remained, but with attenuation for those deemed physically active [18] (see later in this paper for further consideration on this topic). In the following six years, several meta-analyses confirmed these observational findings [19,20,21], and laboratory experimental evidence linking sedentary behaviour with biomarkers of chronic disease risk (e.g., 2-hour fasting glucose) emerged [22,23]. The latest U.S. guidelines state that “the potential population health impact of sedentary behavior is substantial” [24]. Numerous studies of the correlates of sedentary behaviour have also emerged [25], and subsequent intervention trials have been developed [26] and scaled-up for translation to larger numbers [27]. Moreover, the first two edited books have recently been published, specifically focussing on sedentary behaviour and health, covering more than 50 chapters and 1,000 pages [28,29].

With such rapid change in this research field over only the past decade or so, it is inevitable that evidence and views will develop to become more advanced, diverse and, possibly, contentious [30]. This is no different from the physical activity literature or other areas of health behaviour research, such as nutrition research, where an evolution of thinking is evident and inevitable. Indeed, researchers in psychology have labelled the changing landscape in a new area of research the ‘decline effect’ [31]. This does not mean that evidence necessarily gets less supportive of an initial point of view, but rather that evidence builds in a more sophisticated and nuanced way. Studies get larger, use better measures, control for important confounders in better ways and, where feasible, provide experimental rather than just observational evidence. It is almost inevitable that early, possibly simplistic, conclusions will be refined. Some hypotheses will continue to be supported, others modified. 

For physical activity, for example, early guidelines stated that ‘exercise’ needed to be undertaken on a few days of the week largely at a vigorous intensity [32]. Evidence later suggested that substantial benefits for health could be gained by moderate intensity and moderate-to-vigorous intensity physical activity undertaken more frequently [33]. The initial guideline was not incorrect, but given the emergence of newer evidence, it was modified over time. Indeed, physical activity guidelines are still being changed and clarified [34]. Similarly, early research on the ‘causal’ association between TV viewing and adiposity in children [9] was later modified to a more cautious conclusion that such an association, while evident, is complex and likely dependent on many factors [35]. In short, we should not expect a field of research to stand still, but to evolve, develop and become more sophisticated in its evidence and conclusions. Given the rapidly developing evidence base for sedentary behaviour, therefore, some areas for consideration and debate have emerged. These include the relative effects of standing rather than sitting, the contribution of patterns of sitting accumulation rather than just total time sitting and the role of MVPA in modifying associations between sitting and health. These issues formed the basis for the Think Tank reported in this paper.

The ‘Queensland Sedentary Behaviour Think Tank’ meeting was held in May 2018 and brought together people with extensive expertise on sedentary behaviour in the state of Queensland, Australia. The primary purpose was to discuss areas of debate that have emerged within the sedentary behaviour literature, primarily concerning adults. 

## 2. Methods

### 2.1. Participants

An invitation to attend the Think Tank was initially extended to researchers within Queensland, Australia; however, we were fortunate that additional physical activity and sedentary behaviour experts were willing and able to attend the meeting (including one by video conferencing). 

### 2.2. Procedures

The purpose of the meeting was to address three key issues that have been central to early research on sedentary behaviour and health and have recently garnered debate: the health effects of standing [30], the role of targeting prolonged sedentary time (‘sedentary breaks’) [36] and the interactive effects of MVPA and sedentary behaviour on health outcomes [37,38]. The meeting invited three experts to lead 10-minute ‘stimulus presentations’ covering each of these topics. Another expert chaired the session, consisting of a discussion of about 50 min, and a third person took notes alongside an audio recording (see below). 

#### 2.2.1. Stimulus Presentations

These 10-minute presentations by invited experts were not comprehensive systematic reviews of evidence. Rather, ‘expert opinion’ was used in the presentation of evidence from key reviews and large-scale studies, drawing conclusions that were then subjected to scrutiny and discussion. The summary content from these presentations is presented in this paper. Only in exceptional cases did we include important updated evidence subsequent to the Think Tank when writing this manuscript. 

#### 2.2.2. Topic-Focussed Discussions 

During and after each presentation, discussions were recorded and transcribed. As a backup, notes were taken by PhD students (one student per topic area). The discussion content was analysed by the first and third authors using thematic narrative analysis [39].

The meeting was designed as a ‘think tank’. This is a form of meeting where the agenda is open to members to discuss evidence, perspectives and future directions with a view to clarifying inconsistencies and highlighting priority areas of work.

Prior to coverage of each of the three selected topics, participants were reminded of the complexity of the behaviour in question. In any appraisal of health outcomes of sedentary behaviour, we should note the following:Sedentary behaviour has been shown to have differential effects on some health outcomes. For example, vascular and metabolic outcomes may show stronger effects than weight loss or mood and may vary by initial health status. However, despite being a comparatively weaker, albeit opposing, physiological stimulus (vs. moderate-vigorous physical activity), sedentary behaviour is often undertaken for long periods and is highly prevalent in many population groups. Regular exercise has clear effects on multiple aspects of health (i.e., can be a strong stimulus) but is undertaken for short periods and has low population prevalence.The behaviour itself—sedentary behaviour—is commonly assessed through self-reported methods or with wearable technology. Measures typical in the research literature include total sedentary time or time spent in individual domains of behaviour or settings, such as screen-viewing time or sitting at work.Assessment of health-related outcomes have included mortality, incident chronic disease (e.g., colon cancer, type II diabetes), cardio-metabolic biomarkers, adiposity, musculo-skeletal integrity, physical fitness, physical function, mental health and behavioural outcomes (e.g., work engagement).Combining all the above factors makes the area complex, and one should not necessarily expect simple or single answers or solutions.

## 3. Results

A total of 21 researchers were invited to participate in the Think Tank, including 15 from Queensland, three from other states of Australia and two from overseas (Sweden and U.S.). Most participants were university employees (n = 17), with two from medical research institutes and one based at a not-for-profit health agency. One PhD student was invited specifically for her expertise and is included in the 21, with three additional PhD candidates attending and assisting, with the option of contributing to the discussion (not included in the 21).

### 3.1. Topic 1: Is Standing a Sufficient Stimulus to Mitigate the Detrimental Health Effects of Prolonged Sitting?

A narrative review process examining the current literature on the health effects of standing included 11 review studies (n = 6 on the health effects of sit-stand workstations), two prospective studies, nine isotemporal substitution analyses and several experimental studies (see Table 1 for overview).

#### 3.1.1. Review-Level Evidence

The majority of the review studies looking at the association of standing and health outcomes have focused on the impacts of prolonged, static standing. Most evidence is on musculoskeletal outcomes among healthy populations. Occupational (prolonged) standing was found to be associated with increased risks of low back pain [40,41], vascular problems [40], fatigue and discomfort [40] and lower extremity symptoms [41], but not with upper extremity symptoms [41]. While causality between occupational standing and low back pain could not be shown [42], a dose-response analysis of laboratory studies reported clinically relevant levels of low back symptoms after (i) 71 min of prolonged standing in the general population and (ii) 42 min in those considered ‘pain developers’ [43].

There has been limited review-level evidence reporting the potential benefits of standing compared with sitting. A recent meta-analysis showed that energy expenditure is modestly higher when standing compared with sitting [44].

#### 3.1.2. Prospective Evidence

Large cohort studies have examined prospective associations between self-reported standing and mortality. In Canadian adults, a negative dose-response relationship was found between standing (over a quarter of the time) with all-cause, CVD and other causes of mortality [45], but not with cancer mortality. These associations (greater amounts of standing related to lower mortality risks) were true for both men and women but only among physically inactive individuals (see Topic 3). Similarly, Australian data for adults aged 45 years and older also revealed a beneficial relation between standing (over 2 h/day) and all-cause mortality. This association was found to be consistent across subgroups based on sex, BMI, sitting time, physical activity and cardiovascular and diabetes health status [46]. In contrast, Smith et al. [47] reported that people in occupations involving predominantly standing had an approximately 2-fold increased risk of heart disease compared with occupations involving predominantly sitting.

#### 3.1.3. Isotemporal Substitution

Across a 24-hour day, movement behaviours are finite. That is, by not being physically active, for example, one must be either sedentary or sleeping. To test for the hypothetical effects of such behavioural substitution, isotemporal substitution analysis can be used. This is a statistical method that allows for the estimation of the effect of replacing one behaviour (e.g., physical activity) with another (e.g., sitting). 

Overall, studies using isotemporal substitution have estimated that when 30–60 min of sitting is replaced with standing, favourable effects may be observed for cardio-metabolic health [48,49,50,51], mortality [52], metabolic syndrome [51], type 2 diabetes [51], inflammation [53], physical functioning [54], disability [54] and fatigue [54]. However, no favourable effects were observed for cardiorespiratory fitness [55], quality of life [54], role and social functioning [54], depression [54] or anxiety [54]. The evidence for reduced adiposity after replacing sitting with standing was inconsistent [49,51,56].

#### 3.1.4. Experimental Evidence

Experimental studies, mostly in smaller samples (n = 10–20) of healthy or overweight/obese adults, have examined the acute (typically 1 day) effects of increased standing on health outcomes. Inconsistent findings were observed regarding cardio-metabolic outcomes [57,58,59,60], blood pressure [61,62], musculoskeletal outcomes [63,64,65], fatigue [65], mood [59] and cognitive function [59,65,66,67,68]. However, energy expenditure was shown to be increased in the afternoon standing experimental conditions, by 174 kcal over a 210-minute period. By contrast, the meta-analytic review by Saeidifard et al. [44] found a more modest effect, with a mean difference of 0.15 kcal/minute between standing and sitting. 

#### 3.1.5. Sit-to-Stand Workstations

Regarding the use of sit-to-stand workstations, reviews showed mixed findings regarding kinematics [69], physiological health [69,70], (low back/ musculoskeletal) discomfort [69,71,72,73], mood states [70] and energy expenditure [70,74]. Productivity/performance [70,71,73,74,75] and sick leave [73] were not decreased by the use of sit-to-stand desks.

#### 3.1.6. Research Summary 

Some review studies show that energy expenditure is marginally higher when standing compared with sitting. However, prolonged standing is positively associated with low back pain, lower extremity symptoms and fatigue. Currently, there is a lack of meta-analyses or review studies looking at the link between (prolonged) standing and mortality, cardio-metabolic and cognitive outcomes. However, two prospective cohort studies suggest a negative dose-response relation with mortality. Lastly, studies based on isotemporal substitution analyses seem to support the benefits of replacing sitting with standing for some (cardio-metabolic) health outcomes.

#### 3.1.7. Think Tank Discussion

Initial discussion among the experts recognised the importance of early relevant physiological research on more extreme forms of sitting/standing through studies on bed rest and space flight weightlessness. Analysis of the transcript of the discussion revealed two main themes, in addition to future directions. The two themes concern the measurement of standing and the messaging of standing/sitting for guidelines.

Measurement of standing.

Three sub-themes were evident from the discussion on measurement. The first concerned measurement methods, the second referred to patterns of standing behaviours and the third was on the context of assessing standing. Regarding methods, it was considered important to agree to a definition of standing and to differentiate passive from active standing. Tremblay et al. [1] p. 9, define standing as “a position in which one has or is maintaining an upright position while supported by one’s feet”, with passive standing defined as a “standing posture characterized by an energy expenditure ≤2.0 METs, while standing without ambulation” and active standing defined as “a standing posture characterized by an energy expenditure >2.0 METs, while standing without ambulation”.

Debate among the experts in the Think Tank centered on whether we need to develop measures of standing, for example, in cohort studies, or whether assessing sitting was sufficient. The difficulties in differentiating forms of standing from light-intensity physical activity (LIPA) were noted. Moreover, it may be sufficient, and indeed preferential, to employ gold standard measures of posture (e.g., with inclinometers and accelerometers) than self-reported measures of standing/sitting. As with physical activity and sitting measurements, the quality of self-reported measures of standing was questioned, although estimating the percentage of time spent standing in some contexts (e.g., work) might be useful [16].

In the second sub-theme, it was agreed that very little is known about patterns of standing in terms of accumulation or short and long bouts of daily standing. For example, cohort studies that have used device-based measures, such as AusDiab [49], suggest that people stand for about 4 h/day, but little is known about the patterning of this. Similarly, people may report that they have ‘been on their feet’ for large portions of the day, but the validity of such statements is not known. 

The third sub-theme concerned the context of standing. It was noted that there will be variability within occupations for standing, and related issues involved the free will to stand and the social and behavioural contexts of standing. For example, Mansfield et al. [76] reported that some people are reluctant to stand in some work-related contexts as they felt it was a form of ‘norm violation’. 

Evidence to inform messages on standing.

It was noted that, with the exception of the statement by Buckley and colleagues [77], there is little said in guideline documents about standing. It was noted that the evidence-informed determination of any messaging concerning standing would be a complex matter. Some of the key issues in this complexity are as follows: too much (prolonged) standing is assumed to be harmful; any health outcomes of standing will depend on the outcome of interest (e.g., musculoskeletal vs. metabolic); a simple energy expenditure explanation for any health effects of standing is too simplistic; the context will vary greatly—not everyone is at work, and some occupations will involve high levels of standing; there will be large differences in contexts between young people, adults and older adults; psychological reactions to standing will partly be dependent on the degree of free will involved; some forms of standing will be more dynamic than others.

The experts agreed that any evidence-informed messages concerning standing should be focused on the importance of frequent postural changes and breaking up prolonged sitting as often as possible.

Key future research directions.

While many of the issues raised so far can be formulated as research questions, additional key research directions identified by the experts included the identification of the correlates and determinants of standing, consistent with the behavioural epidemiological framework: the identification of the facilitators of standing; better understanding of dose-response effects of standing, including volume, time and patterns; and examination of underlying mechanisms of positive and negative effects of different bouts of standing.

### 3.2. Topic 2: Are there Beneficial Effects for Breaking up Sedentary Time?

The importance of reducing the duration of prolonged sitting bouts (i.e., breaking up sitting) has been accepted in many statements from national guidelines. For example, guidance in Australia uses the phrase “Minimise the amount of time spent in prolonged sitting” (see http://www.health.gov.au/internet/main/publishing.nsf/content/health-pubhlth-strateg-phys-act-guidelines#apaadult). It has long been thought that prolonged sitting is not healthy, as shown in bed rest studies, with astronauts during weightless space flights [78] and commonly held assumptions regarding breaking up sitting on long-haul commercial flights. However, it was not until 2008, after several years of research on sedentary behaviour, that researchers presented data suggesting breaking up prolonged sitting might have important effects for health [79].

#### 3.2.1. Observational and Isotemporal Substitution Studies

The first study to investigate the effects of breaks in sedentary time comprised a sample of 168 adults recruited from the Australian Diabetes, Obesity and Lifestyle (AusDiab) Study [79]. Hip-worn accelerometer-determined sedentary time was assessed over one week. Data were analysed in 60 s epochs, with a break in sedentary time identified as a transition between a sedentary (<100 counts per minute) and non-sedentary epoch. The total number of breaks from sedentary time was beneficially associated with metabolic markers, especially for lower waist circumference and BMI, lower triglycerides and lower 2-h plasma glucose. These effects were shown after controlling for total sedentary time and MVPA. 

Further tests of associations with sedentary breaks were made through the 2003/04 and 2005/06 US National Health and Nutrition Examination Survey (NHANES) [80]. Cross-sectional analyses were conducted using data from more than 4700 participants using accelerometer-assessed sedentary time and breaks in sedentary time. Findings were in line with those from the AusDiab study, with breaks in sedentary time being beneficially associated with waist circumference and C-reactive protein (an inflammatory marker) following adjustment for sedentary time and potential confounders. 

A systematic review and meta-analysis by Chastin et al. [81] found that, in observational studies, the only consistent beneficial association for breaks in sedentary time, once the analysis adjusted for total sedentary time, was with indices of obesity. This review also noted that the “breaks” measure (number of transitions from sedentary to non-sedentary) is a simplistic measure of the underlying concept—namely, that how sedentary time is accrued may be relevant for health. Bellettiere et al. [82] used the activPAL data collected in the 3rd wave of the AusDiab study to explore this concept in more depth, examining the associations of sitting time, sitting time accrued in prolonged bouts ≥30 min and three measures of sitting accumulation patterns, together with cardio-metabolic risk markers. They reported that sitting accumulation patterns that reflected more frequent interruptions (compared with those with relatively more prolonged sitting) were beneficial for several biomarkers, and the effect sizes for the associations were typically larger for the accumulation patterns than for sitting time volume. Diaz et al. [83] reported that both total sedentary time and the duration of the sedentary bouts were associated with all-cause mortality, with those classified as high for both sedentary time and high bout duration having the highest mortality risk.

#### 3.2.2. Experimental and Review-Level Evidence

Building on observational evidence, tests of the effects of sedentary breaks on health indicators have been undertaken in experimental settings. However, it should be noted that in observational research, sedentary breaks have typically reflected an interruption (i.e., transition from sitting to standing or moving) as determined by accelerometer cut-points. On the other hand, the focus in the experimental trials has been on investigating ‘activity breaks’ during prolonged sitting whereby periods of sitting have been intermittently replaced by some form of physical activity, ranging from standing to moderate-to-vigorous activity. Typically, the prolonged sitting condition in the experimental studies has been used as the ‘control’, and the mitigating influence of the activity breaks has been the focus of attention. Recent pooled analyses of laboratory-based trials indicate that higher energy expenditures of different types of activity breaks (standing, light- or moderate-intensity walking) were associated with lower postprandial glucose and insulin responses in a dose-response manner in overweight/obese sedentary adults [84], and that those with higher underlying levels of insulin resistance may derive greater metabolic benefits from regularly interrupting prolonged sitting with activity breaks than their healthier counterparts [85]. Similar findings were observed in a systematic review and meta-analysis by Saunders et al. [86]. This review compared the acute (<24 h) effects of prolonged sitting with those of repeated short bouts of light to moderate activity (‘regular activity breaks’, less than 10 min in duration) on postprandial glucose, insulin and triglyceride concentrations. Prolonged sitting resulted in higher postprandial glucose and insulin than when light- or moderate-intensity physical activity breaks interrupted the sitting. The medium effect size was considered to be “clinically relevant if experienced on a regular basis” [85] p. 2352. However, these effects seemed more evident for light- and moderate-intensity movement rather than standing per se. The authors of this review concluded that standing breaks that are less than about 10 min may not be sufficient to change some of these cardiometabolic biomarkers, especially in healthy participants.

Duvivier et al. [87] provided preliminary experimental evidence that the patterning of the interruptions is important, reporting that when energy expenditure is comparable, standing and walking for longer duration improved cardio-metabolic biomarkers more so than shorter periods of higher intensity physical activity, at least in the short term.

#### 3.2.3. Research Summary

Bed rest studies have clearly demonstrated that prolonged, unbroken sitting is harmful; however, the sophistication of measures needed to assess sedentary accumulation patterns in free-living adults means that this research area is in its infancy. Nevertheless, both observational and experimental studies suggest that the pattern of sitting accumulation may be important for messages concerning the reduction of sitting time.

#### 3.2.4. Think Tank Discussion

The experts agreed that the desirability of breaking up prolonged bouts of sitting was supported by empirical and anecdotal evidence. The converse was not to oppose nor discourage prolonged sitting. The discussion centered on the frequency of breaks and the messaging of sitting breaks.

Frequency of breaks from sitting.

It was recognised that differences in biomarker assessments when analysed by the frequency of sitting breaks are small (but still important) compared with those from moderate or higher intensity physical activity. Experts agreed that it was appropriate to encourage more breaks from sitting and more physical activity. The importance of the smaller stimulus of breaking up prolonged sitting, however, is thought to be particularly significant for those with chronic disease and for older adults. For example, rising from a seated position requires strength and balance, both important markers of physical function. In the latest U.S. guidelines, however, it was stated that “the literature was insufficient to recommend a specific target for adults or youth for how many times during the day sedentary time should be interrupted with physical activity” [24] p. 21.

Informing messages about breaks in sitting.

The expert group was very clear: it is important to encourage less sedentary behaviour and more physical activity. It should not be stated as ‘or’. It was felt that experts needed to be cautious about being too prescriptive in light of the current state of evidence. However, in the context of advice on evidence-informed messages, sometimes it was felt that a focus on time limits could be justified, despite a lack of strong evidence on what these times should be. Based on other occupational and behavioural guidelines, as well as expert opinion, an emphasis on breaking up sitting with movement every 30 min was thought likely to be appropriate. Moreover, it was felt that there would be no risk or down-side to this; it would be encouraging the breaking up of sitting and change of posture. These issues are more behavioural than biological, and account must be taken of preferences and circumstances. Discussion at the meeting also considered taking standing breaks (from sitting) and felt that these would be acceptable, whereas messages about sitting as little as possible may be less likely to be accepted [88]. Overall, there was clear consensus about the importance of breaking up prolonged sitting time, even if it is not yet known with quantitative precision how often or for how long to do this.

Future research directions.

Three priority research issues were identified. First, there is a need to better understand how people will react to different messages concerning sitting breaks, including across different groups that vary in physical function, age and chronic disease. Second, different study designs may be required to address different research questions, such as adoption and maintenance of the behaviour of breaking up prolonged sitting. Third, more research is needed on identifying the mechanisms of any benefits of breaking up prolonged sitting; examples include blood flow and vascular reaction.

### 3.3. Topic 3: Does Moderate-to-Vigorous Physical Activity Attenuate the Adverse Health Effects of Sitting?

Consideration was given to literature examining the adverse health effects of sedentary behaviour when either adjusting for the effects of MVPA or when different levels of MVPA were used to test for moderation effects on the associations between sedentary behaviour and health outcomes. Results were drawn from systematic reviews and cohort studies. To make the process manageable, mortality was used as the primary health outcome marker. 

#### 3.3.1. Effects of Sitting or Sedentary Behaviour after Adjusting for the Effects of Physical Activity

Initial studies tended to adjust statistical models for time spent in MVPA—usually in leisure time—when estimating the effects of sedentary behaviour on health. For example, the European Prospective Investigation of Cancer (EPIC) Norfolk cohort study in the UK reported by Wijndaele et al. [89] showed that TV viewing time was associated with mortality (all-cause and cardiovascular, but not cancer) after controlling for confounders that included physical activity energy expenditure. With a growing number of similar studies assessing these associations, along with various measures of sedentary time and mortality, several systematic reviews and meta-analyses have appeared in the literature [90]. For example, Wilmot et al. [19] reviewed eight large cohort studies and compared mortality between the highest and lowest categories of sedentary behaviour. Results showed that the most sedentary were at 49% greater risk of all-cause mortality, and that such associations remained after controlling for MVPA (i.e., adding MVPA to statistical models as a confounder). Based on a review of eight systematic reviews, including the review by Wilmot et al. [19], and an analysis of causality using epidemiological criteria proposed by Hill [91], Biddle et al. [90] concluded that “there is reasonable evidence for a likely causal relationship between sedentary behaviour and all-cause mortality based on the epidemiological criteria of strength of association, consistency of effect, and temporality” [89] p.1. The largest review included in this analysis was by Biswas et al. [20], who analysed only studies that controlled for MVPA in assessing relationships between sedentary behaviour and mortality. They concluded that an association existed “regardless of physical activity” [20] p. 123, but also stated that “the deleterious outcome effects associated with sedentary time generally decreased in magnitude among persons who participated in higher levels of physical activity compared with lower levels” [20] p. 127 (see later).

Since the synthesis of reviews by Biddle et al. [90], there have been two large meta-analyses examining the association between sedentary behaviour and mortality, where physical activity has been considered a confounder. Patterson et al. [92] reported from 12 studies that associations between largely self-reported total sedentary time and all-cause mortality yielded a relative risk (RR) of 1.03 per hour/day. This remained significant and was only marginally attenuated by adjusting for physical activity (RR = 1.02). Such associations were non-linear and suggested that the risk was higher after about 8 h of sedentary time per day. Similar trends were found for TV viewing time, although relative risk values were higher (see Figure 2).

In an analysis of 19 studies and more than one million participants, Ku et al. [93] reported a log-linear association between sedentary time—measured with devices (mainly Actigraph accelerometers) and self-reporting—and all-cause mortality. Inspection of the dose-response curve suggested that a significant risk for all-cause mortality was evident from about 7.5 h of sedentary time per day. However, for self-reported studies, this was around 7 h, while for studies using devices, the value was closer to 9 h. The authors stated that MVPA was controlled for in all studies. In contrast, when using tri-axial accelerometers, the large Women’s Health Study cohort (n = 16,741; mean age at baseline = 72 years) showed that associations between sedentary time and all-cause mortality that were evident in the highest two quartiles of sedentary time were completely attenuated after accounting for time in MVPA [94]. However, Belletiere et al. [95] recently reported on accelerometer data on older women from the OPACH Study (Objective Physical Activity and Cardiovascular Health; n = 5638, aged 63–97 years). This study, with 5-year follow-up, showed that high sedentary time and long mean sedentary bout durations were associated, after adjustment for accelerometer-determined MVPA, in a dose-response manner with increased risk of cardiovascular disease.

In appraising studies that assessed the association between sedentary behaviour and health and where physical activity was adjusted for as a confounder, it could be concluded that sitting time is usually associated with deleterious health outcomes, at least for mortality and mostly for self-reported data. However, this does not answer the question whether those undertaking higher levels of physical activity are protected from negative health outcomes of sitting. Therefore, we also considered studies that analysed MVPA as a moderator of the association between sedentary behaviour and health.

#### 3.3.2. Moderation Effects of MVPA on Sedentary Behaviour and Health Relationships

The first cohort study to consider this was an analysis of the Canada Fitness Survey by Katzmarzyk et al. [18]. Participants were requested to estimate their sitting using a fairly crude instrument comprising five categories ranging from sitting ‘almost none of the time’ to ‘almost all of the time’. Sitting time was positively associated with all-cause, cardiovascular and ‘other’ mortality, but not cancer. When participants were split into ‘inactive’ and ‘active’, the latter being active at a level of at least 7.5 MET h/week—commensurate with current physical activity guidelines—both groups showed a dose-response association between sitting and mortality. However, the effect was attenuated for those deemed physically active such that those most at risk were those who were physically inactive in the two highest sitting groups. 

Using accelerometer data from the National Health and Nutrition Examination Survey (NHANES) in the United States, Loprinzi et al. [96] found that all-cause mortality was associated with higher sedentary time and lower levels of MVPA. However, an increase in sedentary time was not associated with mortality for those above the median for MVPA. 

Ekelund et al. [37] reported results from a harmonised data analysis of studies that examined self-reported sitting time (k = 13), TV viewing time (k = 6) and all-cause mortality. The sitting time studies included 1,005,791 individuals who were followed for 2–18 years. During this period, 84,609 (8.4%) died. Self-reported sitting time was categorised into four groups (0–<4 h/day; 4–<6 h/day; 6–8 h/day; >8h/day), with self-reported MVPA as quartiles. Analyses then investigated the mortality risk for each combination of sitting and MVPA. This was repeated for TV viewing, where four groups were created (<1 h/day; 1–2h/day; 3–4 h/day; >5 h/day). Results showed dose-response curves for sitting time and mortality; however, these flattened out and became non-significant for the highest activity group. There was little or no effect of sitting among those who reported 60–75 min/day of MVPA. For participants meeting physical activity guidelines at 16 MET-h/week, a dose-response effect for sitting was shown for mortality. Effects for TV viewing were stronger; high TV viewers had elevated risk of mortality even if highly physically active. This study concluded that high levels of MVPA can attenuate or even eliminate the deleterious effects of sitting on mortality, but the effects were less marked for TV. The highest quartile for sitting was set at >8 h/day, which is around the level at which risk is expected to increase. In conclusion, in studies in which different levels of MVPA are considered, the effects of high levels of sitting time on mortality are much less marked when PA levels are high, which, in the paper by Ekelund et al. [37], equated with about 60 min of moderate intensity activity or 30 min of vigorous activity daily. While this may initially seem to be a high target, this level of self-reported physical activity was reported by one quarter of those whose data were included in the meta-analysis. An important consideration here and for future studies is that of ‘plausible’ and ‘theoretical’ risk. In the meta-analysis conducted by Ekelund et al. [37], the analysis by quartiles suggests that 75% of the population is at increased risk of mortality from higher levels of sitting, reflecting a ‘plausible risk’ for sedentary behaviour. Depending on the intensity of physical activity, in the U.S. guidelines documentation, Katzmarzyk et al. [97] estimated from self-reported data that 11–33% of the U.S. population would meet the higher level of physical activity reported by Ekelund et al. [37], although it is recognised that this may differ across countries.

#### 3.3.3. Research Summary

Sedentary behaviour is associated with all-cause mortality [96]. However, early conclusions that sedentary behaviour affected health ‘independently’ of levels of physical activity were drawn mainly from studies where MVPA or other markers of physical activity were statistically used as confounding variables. Attenuation effects tended to be small, with some exceptions. However, in studies in which results were analysed by different levels of physical activity, hence testing for moderation effects, higher volumes of physical activity were shown to be protective of the effects of extended periods of sitting on mortality. Strong attenuation effects were evident for mortality when levels of MVPA were high. Hence, it is important to continue to promote increased participation in regular physical activity, particularly for those at the upper end of the range of current U.S. and Australian guidelines, in order to reduce the negative health effects of too much sitting.

#### 3.3.4. Think Tank Discussion

In addition to future research directions, two themes emerged from the discussion: methodological issues and messaging.

Methodological issues.

Three sub-themes were discussed: statistics, exposure and outcome measures and sampling bias. There was agreement that the choice of statistical model was important. There was some disparity between results of studies of sedentary behaviour and mortality when using adjusting or moderating analyses. Given the nature of the relationships between both sedentary time and mortality (‘J’ shaped with an extended flat line at the bottom) and physical activity and mortality (marked decline in risk with any PA, followed by a plateau), this is not surprising. Experts felt that some form of triangulation of findings was needed across study designs and analytic methods, in particular, looking at studies that include multiple assessments of sedentary behaviour in relation to mortality risk.

Regarding exposure and outcome measures, it was noted that the large population cohort studies focussed a great deal on mortality. Synthesis of data on other outcomes needs to expand. Analyses might usefully be conducted by groups of exposure variables (e.g., total vs. domain-specific sedentary behaviours), but with a cautionary note regarding the skewness of the distribution of these variables. With the assessment of sedentary behaviour using accelerometers, it was noted that sedentary time is strongly inversely associated with light-intensity physical activity. Moreover, the clustering of behaviours, such as movement, sleep and diet, makes the field complex. Regarding sampling, concern was expressed about sampling bias in cohort studies and the likelihood that the most sedentary are under-represented.

Messaging concerning the role of MVPA in the health effects of sitting.

Confusion seems to occur when similar results (e.g., effect sizes) have led to different conclusions and messages. This has been the case regarding sitting and obesity where some conclude clear effects and others highlight a complex set of influences [98], as well as messages that over-state an effect (e.g., ‘sitting is the new smoking’) [99]. Moreover, if people think it will be easy to reduce sitting in comparison to being more physically active, the latter may be under-emphasised. This would be a mistake, especially given the known potent effect of MVPA on health and well-being. For this reason, it was restated that messages must include reducing sedentary behaviour and increasing physical activity [100], particularly for those who are already insufficiently active.

Who is responsible for messaging was an important issue debated by the experts. It was agreed that academics must make their research findings accessible through appropriate dissemination, but equally, the pressures for research funding and the need for academic impact may distort messages. The industry stakeholders with a vested interest in either sedentary behaviours (e.g., selling chairs) or reducing sedentary time (e.g., companies selling sit-to-stand desks) may want to highlight certain types of messages and could distort the evidence.

Future research directions.

It was agreed that researchers must work towards a stronger hierarchy of evidence to inform causal inference; triangulation of methods is critical here. In addition, we need to know more about the influence of other health-related behaviours and the optimal mix of concurrent health behaviours. This could include sedentary behaviour, physical activity, sleep and diet. Moreover, associations between sleep, sedentary behavior and different intensities of physical activity require continued investigation across finite periods (e.g., 24-hours) whereby one behavior has to be replaced with another, as recognised in isotemporal substitution and compositional analytic methods. 

## 4. Conclusions and Recommendations

The ‘Queensland Sedentary Behaviour Think Tank’ meeting brought together people with extensive expertise on sedentary behaviour, although it is recognised that they may not represent all perspectives and beliefs within the field of sedentary behaviour. The primary purpose was to discuss areas of debate that have emerged in the literature. Focussed presentations and discussion took place, although systematic reviews were not undertaken, and this is recognised as a limitation. Key conclusions were:Standing is important. Benefits may accrue from postural shifts and these include metabolic and cardiovascular adaptations with considerable health benefits and may or may not be related to energy expenditure.Prolonged (mainly static) standing and prolonged sitting are both bad for health.Postural transitions are vital—‘the best posture is the next posture’; regularly breaking up sitting time and replacing this with postural shifts and movement is important.Many health effects of sitting are evident even after controlling for levels of MVPA, but those undertaking high levels of MVPA are likely to have marked attenuation of deleterious effects of high levels of or prolonged sitting.

For research, it was agreed future directions should include the determinants and facilitators of standing, better knowledge concerning the dose-response effects of standing and an examination of the underlying mechanisms of positive and negative effects of standing. Moreover, research on sitting breaks should address issues concerning how people react to certain types of messages on breaks, how different study designs may be required for adoption and maintenance of sitting breaks and research on the mechanisms of the benefits of breaking up prolonged sitting for wider aspects of health, such as blood flow. Research directions should also address evidence that can lead to conclusions regarding causality, what the influence of other health behaviours might be alongside, or co-existing with, sitting (e.g., diet) and the advancement of the understanding of interrelated behaviours across a 24-h day.

As the field of sedentary behaviour research and translation develops, a number of issues require continued attention, including measurement and messaging. Further work is required concerning the juxtaposition of self-reported domains and contexts of sitting alongside device-based measures, as well as the development and evaluation of evidence-based messages. The latter need to be delivered appropriately for public consumption and without false claims. Moreover, it is important to recognise the complexity of the field and the need for further nuanced approaches and conclusions.

## Figures and Tables

**Figure 1 ijerph-16-04762-f001:**
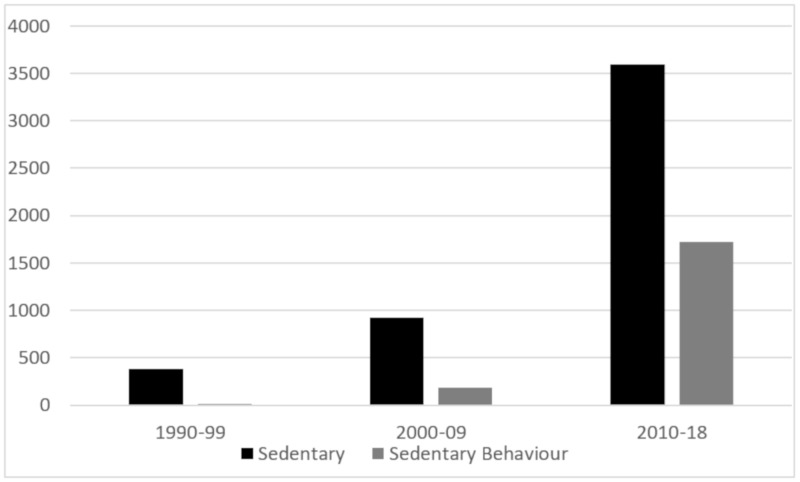
Number of published papers with ‘sedentary’ and ‘sedentary behavio[u]r’ in the title from SCOPUS searches.

**Figure 2 ijerph-16-04762-f002:**
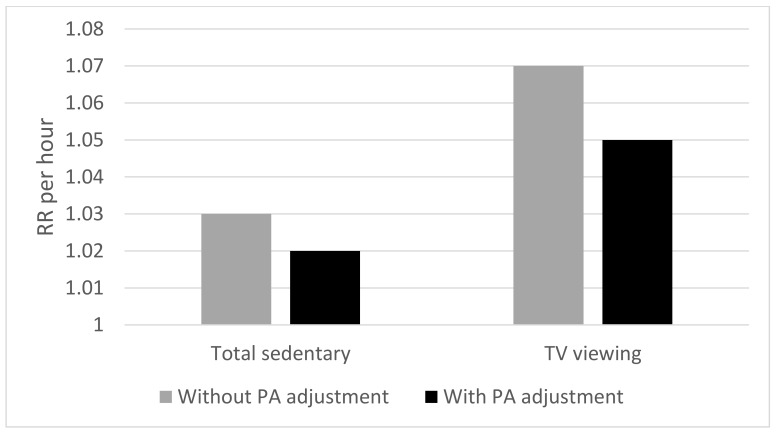
Data from Patterson et al. [91] showing risk ratios per hour for total sedentary behaviour and TV viewing with all-cause mortality.

**Table 1 ijerph-16-04762-t001:** Overview of associations between standing and health outcomes by type of evidence.

Source Type of Evidence	Standing Is in Favour of	Standing Is Detrimental for	Standing Is Not Associated with	Inconsistent Findings on the Link with Standing
Review studies (k = 5)	Energy expenditure (*MA* ^1^) (gender ≠)	Low back pain (*MA* ^1^) Lower extremity symptoms Vascular problems Fatigue & discomfort	Causality low back pain Upper extremity symptoms	
Review studies sit-stand desks (k = 6)			Performance/ productivity Sick leave	Musculoskeletal discomfort Mood states Energy expenditure
Prospective (k = 2)	All-cause mortality CVD ^2^ mortality Other mortality		Cancer mortality	
Isotemporal substitutions (k = 9)	Cardio-metabolic health Mortality Metabolic syndrome Type 2 diabetes Inflammation Physical functioning Disability Fatigue		Cardiorespiratory fitness Quality of life Role functioning Social functioning Depression Anxiety	Adiposity
Experiments (k = 13)	Energy expenditure			Cardio-metabolic health Blood pressure Musculoskeletal health Cognitive functions

^1^ meta-analysis; ^2^ cardio-vascular disease; k = number of studies; ≠: not equal.

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
