# Peer review of "Controversies in the Science of Sedentary Behaviour and Health: Insights, Perspectives and Future Directions from the 2018 Queensland Sedentary Behaviour Think Tank"

_ijerph, 2019, doi:10.3390/ijerph16234762_

Round 1
Reviewer 1 Report
The present paper sums up the discussions from an expert meeting on sedentary behaviour in Queensland 2018. I find the paper well written and reasonably balanced with respect to the discussion regarding the importance of sedentary behaviour and physical activity for health. The paper and the expert opinion is limited by the lack of a systematic literature review. This limitation is clarified up front, but it should also be made clear in the Conclusions and Recommendations section. Otherwise, I have only four minor issues that could be added or clarified.
I believe the inseparable nature of sedentary time and time spent in physical activity could be made clearer. If we assume a 24-h period is finite, as stated in the paper, and we have only two variables, sedentary time and time spent in physical activity, these behaviours will be perfectly negatively correlated. Evidence, for example from isotemporal substitution models, that show replacing sedentary time is favourable for health, does therefore not show sedentary time is detrimental to health. Likewise, experimental evidence must be able to show that the same volume of physical activity have differing effects, when undertaken in different patterns, if removal of/breaks in sedentary behaviors should be proved different from increasing physical activity in general.
In the third section, I believe it is communicated that associations with health for physical activity is stronger than for sedentary behaviour, but this could be stated more clearly. Nevertheless, I support the recommendation to sit less and move more.
The paper is mainly targeting adults and older people, for example through limiting evidence regarding the association to health to mortality outcomes, while the evidence for any detrimental effect of sedentary time is weaker in children. I suggest providing this perspective/limitation up front.
Finally, the recent publication from Ekelund et al, BMJ 2019 is lacking. This paper should be included to inform the discussion.
Reviewer 2 Report
The paper summarises discussion from the Think Tank on sedentary behaviour held in 2018 where three key issues were discussed: 1) the health effects of standing, 2) the role of targeting prolonged sedentary time, and 3) the interactive effects of MVPA and sedentary behaviour on health. The paper is well presenting and relatively easy to follow with a few minor typos. It was encouraging to see qualitative data analysis techniques (thematic narrative analysis) were employed to analyse the discussions.
Abstract
In the title the authors refer to ‘future direction’ and as such I would expect to see a sentence or two in the abstract about this.
Minor
Line 25 – Replace ‘had’ with ‘has’
Line 24 – remove ‘of’
Line 35 – ‘remove ‘that’
Line 42 – remove ‘the’
Line 93-94 – replace ‘also become available’
Line 133 – it would be helpful to know the experts’ areas of expertise if that is possible without compromising anonymity.
Line 196 – replace ‘of’ with ‘with’
Line 522 – replace ‘physically’ with ‘physical’
Major
The expert panel comprised mainly of experts from in and around Queensland, Australia and may not reflect the true breadth of perspectives and beliefs within the field of sedentary behaviour research.
Methods – further information about data analysis should be included such as who conducted the analysis and information about, and reference for the method used.
Conclusion and recommendations – this section would be strengthened by providing clear recommendations for the future direction of sedentary behaviour research in light of the discussions and literature presented in the results section. The authors also suggest sedentary behaviour measurement and messaging is important - a summary of the key issues in the conclusion/recommendations would strengthen the paper.
Line 573-576 – the literature presented in the paper does not support the conclusion that the metabolic and cardiovascular benefits accrued from standing are unrelated to energy expenditure.
